# Tunable entangled photon-pair generation in a liquid crystal

Vitaliy Sultanov[1,2,7], Aljaž Kavčič[3,4,7], Emmanouil Kokkinakis[5], Nerea Sebastián[3], Maria V. Chekhova[1,2 ✉] & Matjaž Humar[3,4,6]

Liquid crystals, with their ability to self-assemble, strong response to an electric field and integrability into complex systems, are key materials in light-beam manipulation[1]. The recently discovered ferroelectric nematic liquid crystals[2,3] also have considerable second-order optical nonlinearity, making them a potential material for nonlinear optics[4,5]. Their use as sources of quantum light could considerably extend the boundaries of photonic quantum technologies[6]. However, spontaneous parametric down-conversion, the basic source of entangled photons[7], heralded single photons[8] and squeezed light[9], has so far not been observed in liquid crystals—or in any liquids or organic materials. Here we implement spontaneous parametric down-conversion in a ferroelectric nematic liquid crystal and demonstrate electric-field tunable broadband generation of entangled photons, with an efficiency comparable to that of the best nonlinear crystals. The emission rate and polarization state of photon pairs is markedly varied by applying a few volts or twisting the molecular orientation along the sample. A liquid-crystal source enables a special type of quasi-phase matching[10], which is based on the molecular twist structure and is therefore reconfigurable for the desired spectral and polarization properties of photon pairs. Such sources promise to outperform standard nonlinear optical materials in terms of functionality, brightness and the tunability of the generated quantum state. The concepts developed here can be extended to complex topological structures, macroscopic devices and multi-pixel tunable quantum light sources.

Liquid crystals (LCs) uniquely combine long-range molecular order and fluidity, which results in the self-assembly of various complex three-dimensional topological structures, birefringence and large response to external stimuli[11]. For this reason, LCs are used in several active optical devices, notably LC displays, tunable filters, spatial light modulators and many others[1,12]. Recently, ferroelectric nematic liquid crystals (FNLCs) have been discovered[2,3,13,14], which have polar ordering, leading to a large dielectric constant, a strong response to an electric field and a very high optical nonlinear response. Among other possible uses, FNLCs have strong potential for applications in tunable nonlinear devices[4,5]. Efficient second-harmonic generation has been demonstrated[4], but the use of LCs as sources of quantum states of light has remained unexplored until now.

Most quantum light sources rely on spontaneous four-wave mixing or spontaneous parametric down-conversion (SPDC). In SPDC, a single photon of a laser source (pump) is converted in a second-order nonlinear material into two daughter photons, which can be entangled in various degrees of freedom. Although SPDC was discovered half a century ago[15–17], and has become the workhorse of quantum optics, sources of photon pairs and heralded photons based on it have barely evolved since then. The necessity to satisfy energy and momentum

conservation laws (also known as the phase-matching condition) implies a careful source design and limits the set of two-photon states that can be generated. Existing solutions such as pump-beam modulation[18], periodically[19] or aperiodically[20] poled nonlinear crystals or waveguides[21], or holographic modulation[22] extend those limits but lack tunability, and are designed for the generation of a particular quantum state. Integrated SPDC sources[23,24] are promising for quantum technologies but still have the same restrictions. The emerging field of quantum-optical metasurfaces[25] pushes the boundaries of quantum-state engineering but at the cost of significantly reduced generation efficiency.

Meanwhile, FNLCs have great potential for quantum optics as a key ingredient for quantum light sources. Changes in the FNLC structure in response to an applied electric field offers fine spatial tuning of the optical properties in real time. Here we demonstrate an electrically tunable source of entangled photons based on SPDC in an LC. To our knowledge, this is the first observation of SPDC in liquid or any organic material; moreover, the SPDC rate is fairly high. We show that the two-photon polarization state can be altered through either a molecular orientation twist along the sample or an applied electric field (Fig. 1a). We believe that this work lays the foundations for an era of tunable quantum light sources.

[1]Friedrich-Alexander Universität Erlangen-Nürnberg, Erlangen, Germany. [2]Max-Planck Institute for the Science of Light, Erlangen, Germany. [3]Jožef Stefan Institute, Ljubljana, Slovenia. [4]Faculty of Mathematics and Physics, University of Ljubljana, Ljubljana, Slovenia. [5]Physics Department, University of Crete, Heraklion, Greece. [6]CENN Nanocenter, Ljubljana, Slovenia. [7]These authors contributed equally: Vitaliy Sultanov, Aljaž Kavčič. ✉e-mail: maria.chekhova@mpl.mpg.de

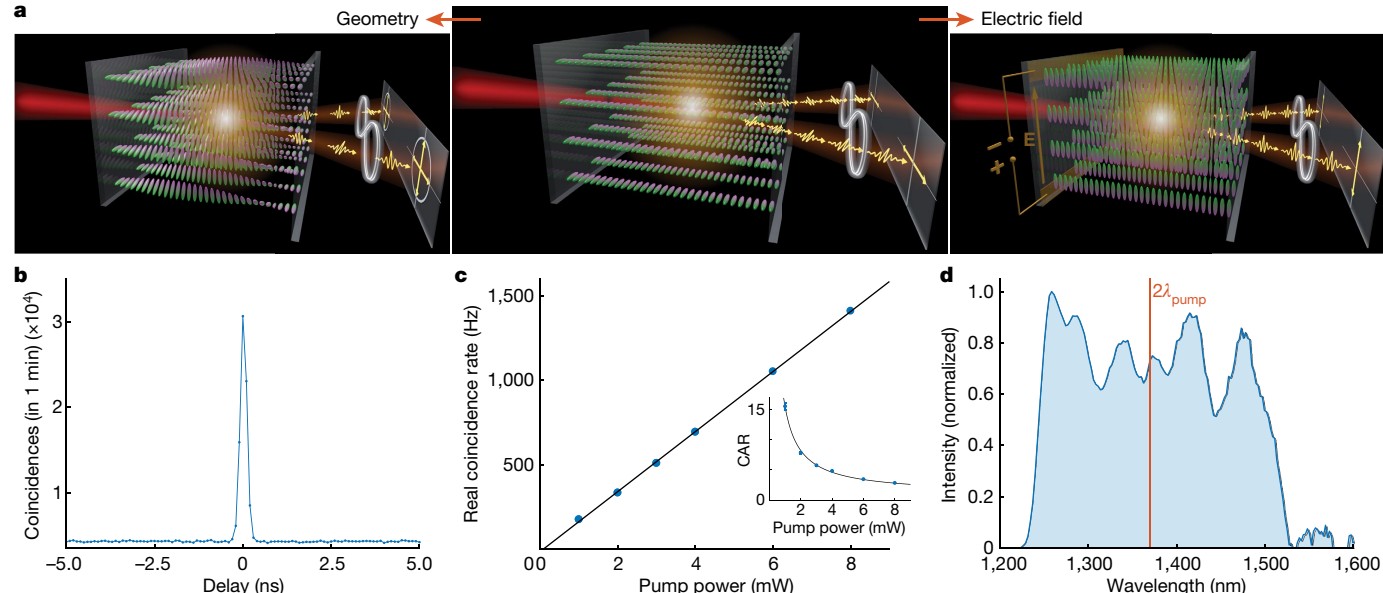

**Fig. 1 | Generation of entangled photons in an FNLC cell. a**, The main concept of the work: both the flux and the polarization state of the photon pairs can be altered by reconfiguring the molecular orientation, achieved by either engineering the sample geometry or applying an electric field. **b**, Typical peak of photon-pair detection (coincidences) at the zero delay between the counts of two detectors. The 8-μm sample has no twist and SPDC is pumped with a 6-mW continuous-wave laser at 685 nm. **c**, The rate of two-photon detection increases linearly with the power of the pump laser. Inset: the contrast of the peak (coincidence-to-accidentals ratio (CAR)) versus the pump power. The inverse dependence indicates two-photon emission. **d**, The spectrum of the generated photon pairs is broadband (limited by the cut-off wavelengths of the used filters) and relatively flat.

## Photon-pair generation in LCs

The nonlinear properties of conventional solid crystals or waveguides used in nonlinear and quantum optics are fixed and defined by the orientation and structure. In contrast, the nonlinearity of FNLCs follows the orientation of the molecules, which can be very complex and can be manipulated by external stimuli. To investigate photon-pair generation in an LC, we prepared several samples with different molecular orientations along the sample, with 0, π/2 or π twist of the molecules (Extended Data Fig. 1). The samples had high values of refractive index and birefringence (Extended Data Fig. 2 and Supplementary Information Section 1) and a thickness of 7–8 μm, which was well below the nonlinear coherence length (about 19 μm), and therefore generated photon pairs coherently[26]. The sample with no twist had two electrodes with a gap of 500 μm between them to create a uniform electric field, which reoriented the molecules (Fig. 1a). Therefore, we separately investigated how a predefined molecular twist and an electric-field-induced change of the molecular orientation affected the state of the generated photon pairs.

Among the three degrees of freedom of two-photon light, such as position/momentum, time/frequency, and polarization, we focus here on polarization because its dependence on the molecules' orientation is the most marked. With the axis $z$ defined along the molecular dipole moment, only one component of the FNLC second-order nonlinear tensor $d_{ij}$ ($i = 1, 2, 3$; $j = 1,...,6$) is significant: $d_{33} \approx 20$ pm V$^{-1}$, only 40% lower than for lithium niobate (Extended Data Fig. 3 and Supplementary Information Section 2). Therefore, photon pairs are generated with the initial polarization along this axis, the orientation of which is effectively equivalent to the orientation of the optic axis of a uniaxial crystal. The interference between all differential polarization states generated along the sample gives the resulting two-photon polarization state.

In each sample, SPDC is pumped with 685-nm continuous-wave laser radiation up to a power of 10 mW focused into a 5-μm spot. At this pump power, we did not observe any other nonlinear effects or permanent damage to the sample (Supplementary Information Section 3). We detect photon pairs with a Hanbury Brown and Twiss interferometer looking at the correlations between the detection times of two photons (Extended Data Fig. 5 and Methods). Photons of the same pair arrive at the detectors simultaneously, creating a peak in the distribution of the time delay between two detection events (Fig. 1b). In contrast, uncorrelated photons from different pairs or generated through a different process (for example, photoluminescence) lead to accidental coincidences, equally distributed over the delay times. Although the number of photon pairs generated via SPDC depends linearly on the pump power (Fig. 1c), the ratio between the height of the peak and the background level of accidental coincidences (coincidence-to-accidental ratio) is inversely proportional to the pump power (Fig. 1c, inset). The data shown in Fig. 1b,c clearly prove photon-pair generation from an LC, with a fairly high coincidence rate. A narrow peak under continuous-wave pumping indicates time-frequency entanglement[26], but we do not quantify it here.

Owing to the microscale source thickness[27] and the resulting relaxed phase-matching condition, the spectrum of photon pairs from an FNLC layer should be broadband. We demonstrate it by measuring the two-photon spectrum (Fig. 1d) via two-photon fibre spectroscopy[28], accounting for the spectral detection efficiency and losses (Extended Data Fig. 6 and Methods). The generated two-photon spectrum is almost flat, up to a modulation caused by the etalon effect inside the source[29], and limited by the frequency filtering. Without filtering, the spectrum of photon pairs is expected to be even broader, suggesting applications such as ultrafast time resolution, high-dimensional time and frequency quantum coding, or hyperentanglement.

## Photon-pair polarization-state switching with electric field

Next, we show how the generated two-photon state changes when the molecules are reoriented under the applied electric field. We measure the rate of real (non-accidental) coincidences for different pump polarizations and polarizations of photon pairs selected with two polarization filters before the detectors. With no field applied, horizontally (H-) oriented molecules generate H-polarized photon

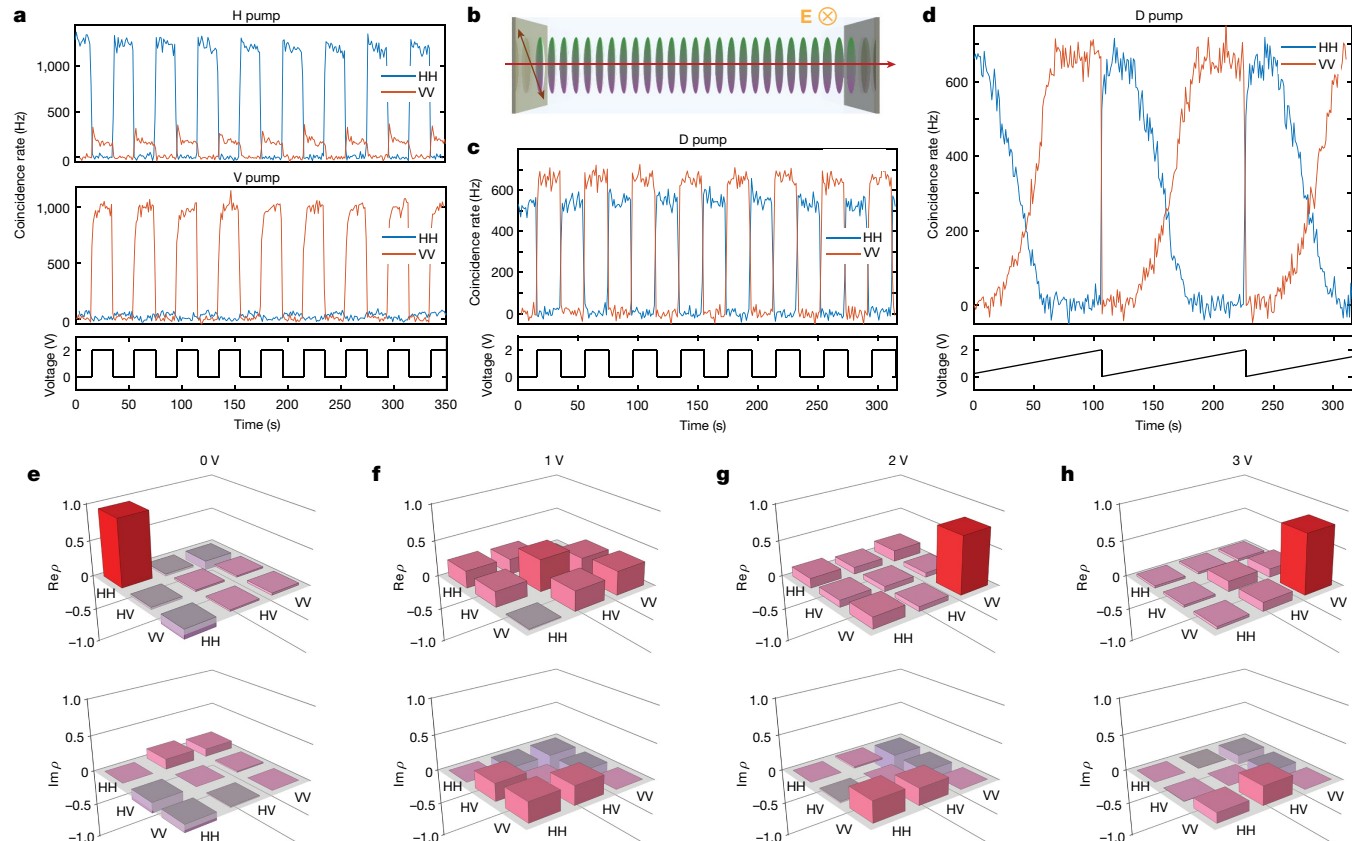

**Fig. 2 | Electric-field tuning of SPDC.** The electric field reorients the molecules, changing both the generation rate and the polarization state of photon pairs. **a**, The rate of pair generation follows the electric-field time dependence and can be modulated significantly for both the horizontally (top) and vertically (bottom) polarized pumps. The H-polarized (blue) and V-polarized (orange) states are turned on and off anti-phased. **b**, The source can be switched between emitting horizontally and vertically polarized states with the D-polarized pump. In the absence of the field, the molecules are oriented horizontally. **c**, The coincidence rate of H- (blue) and V-polarized (red) pairs for the D-polarized pump as a function of the applied voltage. **d**, The rate of pairs varied by applying a sawtooth voltage shows the existence of a threshold voltage required for re-orienting the molecules, and the saturation voltage, where further increase brings no change. **e**–**h**, The polarization state also evolves gradually, according to the reconstructed polarization density matrices under 0 V (**e**), 1 V (**f**), 2 V (**g**) and 3 V (**h**).

pairs from the H-polarized pump and no photon pairs from the vertically (V-) polarized pump (Fig. 2a). Under a field applied perpendicular to the initial molecular orientation, the molecules align with the field (Fig. 1a, middle and right), switching on photon-pair generation. Now, V-polarized photon pairs are generated from the V-polarized pump. The switching happens relatively fast, that is, in about 0.5 s, as measured with second-harmonic generation (Extended Data Fig. 4 and Methods).

The two-photon state switching is even more pronounced when photon pairs are generated with the diagonally (D-) polarized pump (Fig. 2b). The coincidence rate traces are shown in Fig. 2c. It follows that while the generation efficiency is defined by the overlap between the molecular orientation and the pump polarization, the generated state is solely defined by the molecular orientation. The state switching occurs gradually with the increase of the applied field (Fig. 2d), starting at some threshold voltage and reaching the maximum at the saturation point, between 1 V and 2 V, when practically all molecules are oriented along the field. Therefore, a voltage of only 2 V across a 500-µm gap is enough for almost complete switching of the generated two-photon state.

We further investigate the polarization of photon pairs by reconstructing the two-photon state. For classical light or a single photon, the polarization state is a superposition of two basis states, for instance, horizontal and vertical. In contrast, a two-photon state is described by four basis states[30]. However, in the case where two photons are distinguishable in no other way than polarization, the dimensionality is reduced to three, and the state is a qutrit[31,32]

$$|\Psi\rangle = C_1|2\rangle_H|0\rangle_V + C_2|1\rangle_H|1\rangle_V + C_3|0\rangle_H|2\rangle_V, \qquad (1)$$

where $C_1$, $C_2$ and $C_3$ are complex amplitudes, so that $|C_1|^2 + |C_2|^2 + |C_3|^2 = 1$, and $|N\rangle_P$ is a Fock state with $N$ photons in polarization mode $P$. The corresponding density matrix $\rho$ extends the description to the case of mixed states. As we detect photon pairs emitted into the same spatial collinear mode and do not distinguish them in frequencies, we characterize the two-photon state by a three-dimensional density matrix. We reconstruct the two-photon polarization density matrix via polarization tomography[33] optimized via the maximum likelihood method[30]. The details of the experimental procedure can be found in Methods.

As we see in Fig. 2e–h, as the electric field is gradually applied, the two-photon polarization state evolves from both photons polarized horizontally (Fig. 2e) through an intermediate state (Fig. 2f) to the state of both photons polarized vertically (Fig. 2g), which does not change significantly as the field is further increased (Fig. 2h). Therefore, we can obtain either two H- or V-polarized photons or any intermediate two-photon polarization state with the same pump polarization by changing the molecular orientation via an applied electric field.

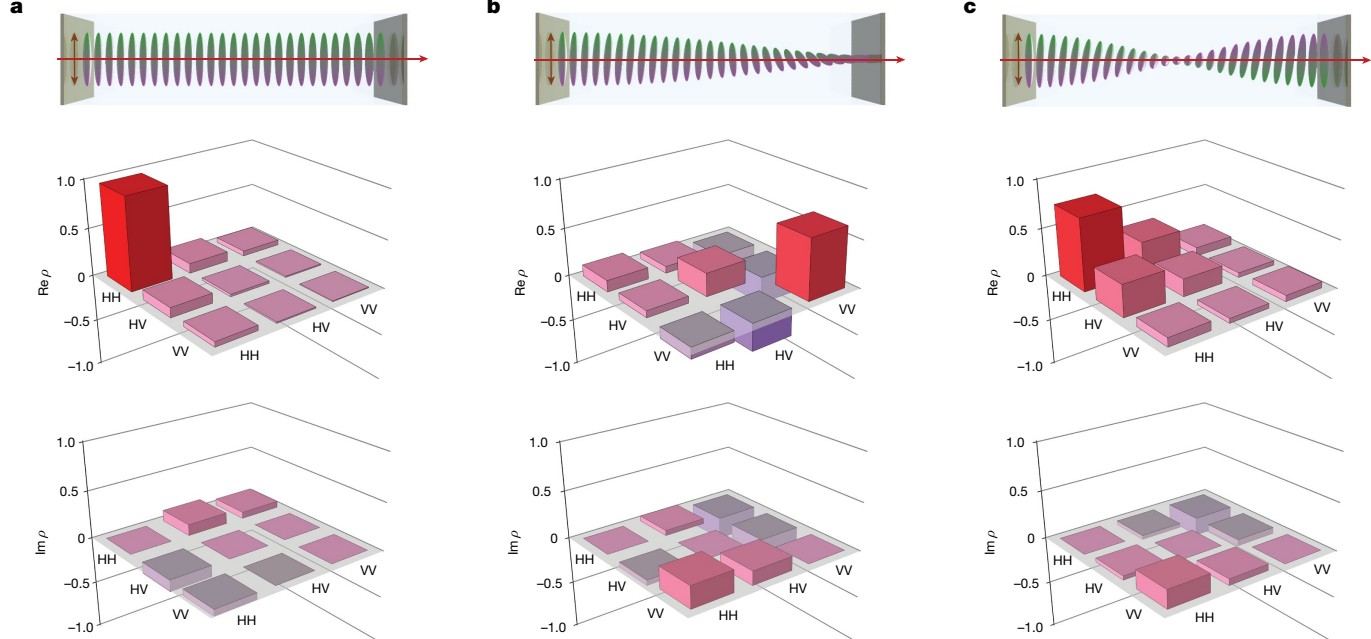

**Fig. 3 | Tuning SPDC by changing the twist geometry.** The density matrix of the two-photon polarization state strongly depends on the twist of the molecules along the sample. **a**–**c**, Examples of 0 (**a**), π/2 (**b**) and π (**c**) twist angles were examined. The horizontal pump-beam polarization coincides with the molecules' axes at the sample's start. Top panels show the molecules' orientation, and the middle and bottom panels show the real and imaginary parts of the measured density matrices, respectively.

## Engineering photon pairs via the sample geometry

Finally, we investigate how the twist of molecular orientation along the sample affects the generated two-photon state (Fig. 3). As expected, an FNLC with no twist and horizontal orientation of the molecules generates pairs of H-polarized photons (Fig. 3a) from a H-polarized pump, according to the nonlinear tensor of the FNLC. In contrast, if molecules gradually change their orientation along the sample from horizontal to vertical (π/2 twist, Fig. 3b), the two-photon state is completely different: it contains mostly 'VV' photon pairs with a small fraction of 'HV' pairs. Further increase of the twist to π twist brings the state close to the H-polarized two-photon state (Fig. 3c). Similar to the applied electric field, the molecular orientation twist gradually changes the state from two photons co-polarized along the molecular orientation (when there is no twist) to an orthogonal state in a superposition with cross-polarized photons.

These results suggest that a broad range of two-photon polarization states can be achieved by engineering the source. To confirm this, we develop a theoretical model (Methods) to investigate the effect of the parameters, such as the FNLC cell length and the molecular twist, on the photon-pair generation. For the FNLC parameters investigated in the experiment (thickness 7–8 µm and twists 0, π/2 and π), our model has fairly good agreement with the experimental results (Fig. 4a). The differences in the calculated and measured polarization states at 90° twist could be attributed to two things. First, the actual twist could slightly differ from 90°, and, second, there is a possibility of slight differences between the linear twist considered for simulations and the actual twist profile of the average molecular orientation across the cell. These concerns were deduced from optical observations (Extended Data Fig. 1).

Further analysis of the parameter space shows that we can achieve two-photon polarization states that, after splitting the pair on a non-polarizing beamsplitter, yield pairs with any given degree of polarization entanglement. The concurrence of such a two-photon state $C = |2C_1C_3 - C_2^2|$, a measure of polarization entanglement[27], spans the whole range of values from 0 to 1 (Fig. 4b).

Moreover, a proper twist of the molecular orientation along the FNLC slab can compensate for the nonlinear phase mismatch[10], as shown in Fig. 4c,d, similar to the periodic poling of a nonlinear crystal. The generation efficiency can be significantly enhanced by properly choosing the twist pitch along a macroscopically thick FNLC. For an optimal twist pitch, in our case $L_{opt} = 19.96$ µm, which is very close to the coherence length 19.25 µm, the rate scales quadratically with the length, same as for SPDC in a phase-matched source. Experimentally, the proper pitch could be easily achieved by doping the FNLC with a chiral dopant[34–36]. In this case, a 200-µm sample will generate 625 times more pairs than the current sample (Fig. 4d), reaching count rates of almost 1 MHz, more than enough for most quantum technological applications. Such performance and the possibility to dynamically control the two-photon state are superior to existing crystal SPDC or fibre spontaneous four-wave mixing sources. The latter are less efficient, require a strong pulsed pump and a relatively long nonlinear medium, and have no polarization tunability without re-designing the source[37,38].

In conclusion, we have demonstrated the successful generation of entangled photons via SPDC in an LC, with an efficiency as high as the most efficient commonly used nonlinear crystals of the same thickness.

One of the most remarkable features discovered in these experiments is the unprecedented tunability of the two-photon state, achieved by manipulating the LC molecular orientation. By re-orienting the molecules through the application of an electric field, we can dynamically switch the polarization state of the generated photon pairs. This level of control over the photon pairs' polarization properties is a crucial advancement, offering opportunities for quantum-state engineering in the sources with pixelwise-tunable optical properties, both linear and nonlinear.

Alternatively, we can manipulate the polarization state by implementing a molecular orientation twist along the sample. This approach adds versatility to the design and utilization of LC-based photon-pair sources. Moreover, a strong twist along the sample can markedly increase the efficiency of a macroscopically large source, similar to the periodic poling of bulk crystals and waveguides, but much simpler

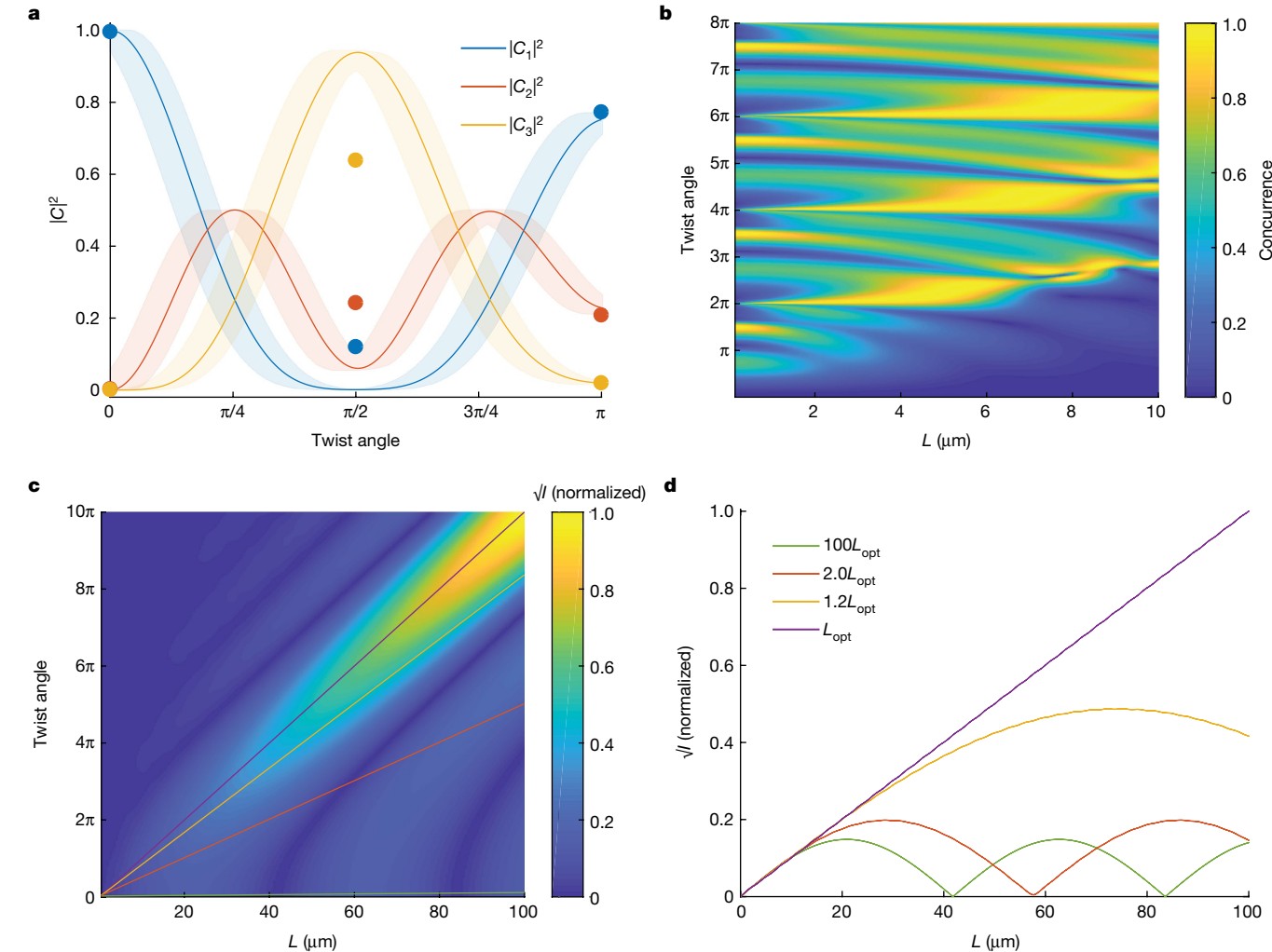

**Fig. 4 | Perspectives of SPDC engineering with molecules' twists.**
**a**, Probabilities of the three-basis two-photon states calculated as functions of the molecules' twist (lines) and measured for the three studied samples (points). The shaded areas around the curves show variation due to fabrication imperfections. **b**, Concurrence of the polarization state calculated versus the sample length and the molecules' twist. **c**, Square root of the pair generation rate $I$ calculated versus the sample length $L$ and molecules' twist along the sample. The straight lines show the cases of different twist pitches. **d**, Dependence of $\sqrt{I}$ on the sample length for various pitches. For an optimal pitch $L_{opt} = 19.96\,\mu m$, the rate scales quadratically with the length, as in perfectly phase-matched crystals.

technologically, as the structure is self-assembled and may be tuned with temperature and electric field[39]. Owing to their nonlinear coefficient comparable to the best nonlinear crystals, such as lithium niobate, and high damage threshold, FNLCs are perfectly suitable for practical applications. Furthermore, high-quality LC devices such as LC displays are made on an industrial scale, which, combined with our work, opens a path to scalable and cheap production of quantum light sources while exceeding the existing ones in efficiency and functionality.

In the future, the electric-field tuning could be expanded to multi-pixel devices, which have the potential to generate tunable high-dimensional entanglement and multiphoton states. Furthermore, FNLCs can self-assemble in a variety of complex topological structures, which are expected to emit photon pairs in complex, spatially varying beams (structured light), such as vector and vortex beams[40]. The liquid nature of FNLCs opens a path to their integration with existing optical platforms such as fibres[41], waveguides[42] and metasurfaces[43].

Overall, the results presented in this paper highlight the potential of LCs for practical applications in quantum technologies. LC-based photon-pair generation with tunable polarization states offers exciting possibilities for quantum information processing, quantum key distribution and quantum-enhanced sensing.

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

## Methods

### Material and sample preparation

The material used in this study is FNLC-1751 supplied by Merck Electronics KGaA. FNLC-1751 shows a stable ferroelectric nematic phase at room temperature, with the phase sequence Iso 87 °C, N 57° C, M2 45 °C, NF on cooling, where Iso refers to the isotropic phase, N to the non-polar nematic phase, M2 to the so-described splay modulated antiferroelectric nematic phase[2,44,45] and NF to the ferroelectric nematic phase.

The material was confined in glass LC cells filled by capillary forces at 100 °C in the isotropic phase. We used both commercially available and homemade cells. In the latter case, soda-lime-glass square 2 cm × 2 cm plates coated with a transparent indium-tin-oxide conductive layer were assembled with plastic bead (EPOSTAR) spacers to achieve a variety of cells with different thicknesses ranging from 7 μm to 8 μm. In the bottom glass, indium-tin-oxide electrodes with a 500-μm gap prepared by etching created the applied in-plane fields. In addition, both plates were treated with a 30% solution of polyimide SUNEVER 5291 (Nissan) film and rubbed to achieve orientational in-plane anchoring (planar alignment) of the LC. Combinations of different relative rubbing directions of the top and bottom glass plates (0, π/2 or π) result in different twist structures of the LC sample in the ferroelectric nematic phase[5,13,46]. In the three cases, the electrode glass was rubbed along the gap. In-plane switching LC cells purchased from Instec were used for switching experiments on π-twisted structures. The cells had interdigitated electrodes in one of the substrates with alternating polarity; both the electrode width and the gap between them were 15 μm. The surfaces had antiparallel rubbing along the electrodes (aligning agent KPI300B).

After filling at 95 °C, the sample was brought to room temperature by controllably decreasing the temperature at a rate of 0.5 °C min⁻¹. In the case of initially large domains breaking down into smaller ones after applying large voltages, the initial configuration was restored by reheating the samples and subsequent controlled cooling back to room temperature. The quality of the achieved alignment was inspected via polarizing optical microscopy and comparison with transmission spectra simulations (Berreman 4 × 4 matrix method performed with the open software package 'dtmm'[47], cell thickness 7.6 μm, and ordinary and extraordinary refractive indices as given in Extended Data Fig. 2).

### Photon-pair generation and detection

The scheme of the experimental set-up used for photon-pair generation and detection is depicted in Extended Data Fig. 5. As a pump, we used a continuous-wave pigtailed single-mode fibre diode laser with a central wavelength of 685 nm. After the power and polarization control, the pump beam was focused into the LC cell with a focusing spot size of 5 μm. The maximum delivered pump power did not exceed 10 mW. A function generator applied to the cell an electric field with different time profiles (Fig. 2). The generated photon pairs were collected with a lens with a numerical aperture of 0.69. Then a set of long-pass filters with a cut-on wavelength no longer than 1,250 nm cut the pump and short-wavelength photoluminescence from the sample and the optical elements of the set-up. Photon pairs were further sent into a Hanbury Brown–Twiss-like (HBT) set-up comprising a non-polarizing beamsplitter and two superconducting nanowire single-photon detectors (SNSPDs). At each output of the non-polarizing beamsplitter, we placed a set of a half-wave plate, a quarter-wave plate and a polarizing beamsplitter, which acted as a polarization filter. The arrival time differences between the pulses of both SNSPDs were registered by a time-tagging device.

### Two-photon spectrum measurement

As the SPDC radiation from an 8-μm layer is extremely weak, direct measurement of the two-photon spectrum (that is, with a spectrometer or optical spectrum analyser) is nearly impossible. Therefore, we measured the spectrum of the detected photon pairs via single-photon fibre spectroscopy[28]. Before one of the SNSPDs, we inserted a 2-km-long dispersion-shifted fibre with a zero-dispersion wavelength at 1.68 μm. Owing to the dispersion of the fibre, the photon wavepacket stretched in time, resulting in a spread of the coincidence peak, which then inherited the spectrum's features and the spectral losses of the set-up. We acquired the coincidence histogram with different sets of spectral filters (Extended Data Fig. 6a) to map the arrival time differences to the corresponding wavelengths of the dispersed photon. The calibration curve (Extended Data Fig. 6b) was obtained by fitting the reference points with a quadratic polynomial function. However, the spectrum is strongly affected by the spectral losses of the set-up and the dispersive fibre. For that reason, we additionally measured the spectrum of photon pairs generated in a thin (7 μm) layer of LiNbO₃ (Extended Data Fig. 6c), where the generated two-photon spectrum is mostly flat, up to a modulation by the Fabry–Pérot effect inside the layer. We then used the spectrum of photon pairs from the LiNbO₃ wafer as a reference spectrum.

### Two-photon-state reconstruction

We performed quantum tomography to reconstruct the two-photon polarization state generated in the LC. The procedure is analogous to measuring the Stokes parameters for classical light or a single photon. By measuring the pair detection rates for different polarization states filtered in the two arms of the HBT set-up, we were able to reconstruct the density matrix of the two-photon state. As there is no prior assumption about the generated two-photon state, we performed all 9 required measurements for the reconstruction of the 3 × 3 density matrix. The full protocol is described in Extended Data Fig. 7. The values in the table refer to the orientation of the fast axis of each wave plate with respect to the horizontal direction. It is worth mentioning that the described protocol does not take into account the mirroring effect of polarization in the reflected arm of the HBT set-up. Therefore, either the angles of the wave plates in the reflected arm must be changed to the opposite values, or an odd number of mirrors must be used in the reflected arm of the HBT set-up. The protocol used for the qutrit state reconstruction is the reduced version of the protocol for the reconstruction of the two-photon polarization state with two distinguishable photons (ququart state)[30].

To avoid systematic errors in the density-matrix reconstruction, we additionally post-processed the measured data using the maximum likelihood method. The maximum likelihood method aims to find the density matrix closest to the measured one that satisfies all basic physical properties of a density matrix. We used a procedure similar to the one described in ref. 30 with minor modifications (Supplementary Information Section 4).

### Theoretical model of SPDC in LCs

We developed a theoretical model to predict the polarization two-photon state generated via SPDC in a nonlinear LC with an arbitrary but linear molecular orientation twist along the cell. The goal is to determine the complex amplitudes of the polarization two-photon state $C_1$, $C_2$ and $C_3$ from equation (1). We assumed a single-mode, collinear and frequency-degenerate photon-pair generation in the plane-wave approximation for simplicity. However, the model can be further extended towards the multi-mode regime of SPDC with realistic angular and frequency spectra, as well as for the case of a non-gradual molecular twist.

Owing to weak interaction, we can use perturbation theory for the unitary transformation of the state vector[48]. The state can be written as

$$|\Psi\rangle = |\text{vac}\rangle + C \int_{-L}^{0} dz \hat{\chi}^{(2)}(z) \; \vdots \; \mathbf{e}_s^*(z)\mathbf{e}_i^*(z)\mathbf{e}_p(z) a_s^\dagger a_i^\dagger |\text{vac}\rangle, \qquad (2)$$

where $a_s^\dagger$ and $a_i^\dagger$ are the photon creation operators for signal and idler photons, respectively, each of them defined in some polarization eigenmode, $\chi^{(2)}(z)$ is the second-order nonlinear tensor, and the polarization vectors $\mathbf{e}_{s,i,p}(z)$ also encode the phase accumulation during the propagation along the crystal of length $L$. Variable $z$ marks the direction of

propagation. The constant $C$ contains only the information about the overall generation efficiency and, therefore, is of no interest to us.

For convenience, we use two polarization bases instead of the polarization eigenmodes. The first basis is a standard linear polarization basis with horizontal and vertical polarizations determined with respect to the laboratory coordinate system, {H, V}. In this basis, the two-photon polarization state can be expressed as a qutrit state (1) as two photons are assumed to be indistinguishable in all other Hilbert spaces apart from polarization[32]. The final goal of the calculations is to determine complex amplitudes $C_{1,2,3}$ from equation (1). As the molecular orientation changes along the crystal and implies the spatial modulation of the nonlinearity, it is more convenient to calculate the convolution of the $\chi^{(2)}$ tensor with the polarization vectors of the interacting photons in the second basis aligned with the instant orientation of the molecules, {e, o}. We denote the corresponding projections with indices $e$ and $o$ for the linear polarization along and orthogonal to the instant molecular orientation, respectively. Instead of the $\chi^{(2)}$ tensor, we use the standard notation of the Kleinman $d$ tensor. Therefore, the convolution is written as

$$\hat{\chi}^{(2)} \vdots \mathbf{e}_s^* \mathbf{e}_i^* \mathbf{e}_p = e_s^{o*} e_i^{o*} (d_{22} e_p^o + d_{32} e_p^e) + e_s^{e*} e_i^{e*} (d_{23} e_p^o + d_{33} e_p^e) + \\ + (e_s^{e*} e_i^{o*} + e_s^{o*} e_i^{e*})(d_{24} e_p^o + d_{34} e_p^e), \tag{3}$$

where the polarization basis vectors and the tensor components are functions of $z$, and the convolution is defined in the local coordinate system of the molecules. The $z$ direction is defined in the same way for both bases and denotes the photon propagation direction along the crystal.

To calculate the polarization two-photon state, we consider an LC with a uniform rotation of the molecules along the crystal (Extended Data Fig. 8). At an arbitrarily chosen layer of thickness d$z$ at position $z$, the pump polarization is modified by all the previous layers it has passed through. The polarization state of photon pairs generated from the corresponding layer d$z$ is further modified by all subsequent layers of the LC. The final state at the output of the crystal is the superposition of all polarizations generated along the crystal. Therefore, to calculate the output two-photon polarization state, we integrate the contribution of each layer of the LC taking into account the corresponding polarization transformations of both the pump and the incremental photon-pair state generated from each layer.

To calculate the propagation of the pump, the initial pump polarization is represented by a Jones vector (Extended Data Fig. 8) in the {H, V} basis. The angle $\varphi_0$ is defined as the angle between both coordinate systems at the beginning of the sample, that is, the angle between the global coordinate H direction and the extraordinary molecule axis $e$ at the beginning of the sample. The first step is to bring the pump from the global basis to the local basis at the beginning of the sample via rotating the pump Jones vector by $\varphi_0$:

$$\mathbf{e}_p^{in} = R(\varphi_0) \, \mathbf{e}_p^o, \tag{4}$$

where $R$ is the standard rotation matrix. The polarization transformation of light propagating through a twisted nematic LC (TLC) with a uniform twist is described by the corresponding Jones matrix[49–51]

$$M_{TLC} = e^{i\phi} R(-\varphi) M(\varphi, \beta);$$

$$M(\varphi, \beta) = \begin{pmatrix} \cos X + i\frac{\beta}{X}\sin X & \frac{\varphi}{X}\sin X \\ -\frac{\varphi}{X}\sin X & \cos X - i\frac{\beta}{X}\sin X \end{pmatrix}. \tag{5}$$

Here, $\phi = \tilde{k} l$ is the average phase acquired by both polarizations, with $\tilde{k} = \frac{1}{2}(k^e + k^o)$ being the average $\mathbf{k}$ vector and $l$ being the thickness of the TLC layer performing polarization transformation; $\varphi$ is the

twist angle; $\beta = \pi l(n_e - n_o)/\lambda = gl$ characterizes birefringence, where we introduce notation $g = \frac{1}{2}(k^e - k^o)$, and $n_o$ and $n_e$ are ordinary and extraordinary refractive indices of the sample at optical wavelength $\lambda$, respectively. The additional parameter $X$ is defined as $X = \sqrt{\varphi^2 + \beta^2}$.

At a certain chosen position $z$, the pump polarization is transformed by the part of the LC from $-L$ to $z$, with the effective length of this layer being $z + L$. The pump polarization vector in the local basis at position $z$ then has the form

$$\begin{pmatrix} e_p^e(z) \\ e_p^o(z) \end{pmatrix} = e^{i\tilde{k}_p(z+L)} M\left(\frac{z+L}{L}\varphi, (z+L)g_p\right) R(\varphi_0) \begin{pmatrix} e_p^H \\ e_p^V \end{pmatrix}, \tag{6}$$

where $\varphi$ denotes the full twist of the sample. We intentionally leave the pump polarization defined in the local basis as it is convenient for calculating its convolution with $\hat{\chi}^{(2)}$. We explicitly write the pump polarization vector at position $z$ in the local basis as a function of the input pump polarization in the {H, V} basis

$$e_p^e(z) = [t_p \, e_p^H + r_p \, e_p^V]e^{i\tilde{k}_p(z+L)}, \\ e_p^o(z) = [t_p^* \, e_p^V - r_p^* \, e_p^H]e^{i\tilde{k}_p(z+L)}, \tag{7}$$

where

$$t_p = \xi_p \cos\varphi_0 - \mu_p \sin\varphi_0, \\ r_p = \xi_p \sin\varphi_0 + \mu_p \cos\varphi_0, \\ \xi_p = \cos\left(\frac{z+L}{L}X_p\right) + i\frac{g_p L}{X_p}\sin\left(\frac{z+L}{L}X_p\right), \\ \mu_p = \frac{\varphi}{X_p}\sin\left(\frac{z+L}{L}X_p\right), \\ X_p = \sqrt{\varphi^2 + (g_p L)^2}. \tag{8}$$

By inserting these expressions into equation (3), we can find the polarization state of photon pairs generated from a unit layer at position $z$ in the local basis. However, as we are interested in the output polarization state, the polarization of both signal and idler photons must be propagated from $z$ to the end of the crystal in a similar way. This transformation can be written as

$$\begin{pmatrix} e_{s,i}^H \\ e_{s,i}^V \end{pmatrix} = R(-\varphi_0 - \varphi)e^{i\tilde{k}_{s,i}(-z)} M\left(\frac{-z}{L}\varphi, -zg_{s,i}\right) \begin{pmatrix} e_{s,i}^e(z) \\ e_{s,i}^o(z) \end{pmatrix}, \tag{9}$$

where the photons are propagating from $z$ to 0. The explicit form of the output polarization for the signal and idler photons generated at $z$ is

$$e_{s,i}^H = [t_{s,i} \, e_{s,i}^e(z) + r_{s,i} \, e_{s,i}^o(z)]e^{-i\tilde{k}_{s,i}z} \\ e_{s,i}^V = [t_{s,i}^* \, e_{s,i}^o(z) - r_{s,i}^* \, e_{s,i}^e(z)]e^{-i\tilde{k}_{s,i}z}, \tag{10}$$

with similar notation as before

$$t_{s,i} = \xi_{s,i}\cos(\varphi_0 + \varphi) + \mu_{s,i}^*\sin(\varphi_0 + \varphi) \\ r_{s,i} = -\xi_{s,i}^*\sin(\varphi_0 + \varphi) + \mu_{s,i}\cos(\varphi_0 + \varphi) \\ \xi_{s,i} = \cos\left(\frac{-z}{L}X_{s,i}\right) + i\frac{g_{s,i}L}{X_{s,i}}\sin\left(\frac{-z}{L}X_{s,i}\right) \\ \mu_{s,i} = \frac{\varphi}{X_{s,i}}\sin\left(\frac{-z}{L}X_{s,i}\right) \\ X_{s,i} = \sqrt{\varphi^2 + (g_{s,i}L)^2}. \tag{11}$$

To perform convolution (3), equation (10) needs to be reversed to express $e_{s,i}^{e,o}(z)$ as functions of the outcome polarizations $e_{s,i}^{H,V}$. With this transformation, alongside equations (3) and (7) the convolution is written as

$$\hat{\chi}^{(2)} : \mathbf{e}_s^* \mathbf{e}_i^* \mathbf{e}_p = [(r_s e_s^{H^*} + t_s^* e_s^{V^*})(r_i e_i^{H^*} + t_i^* e_i^{V^*})P_1$$
$$+ (t_s e_s^{H^*} - r_s^* e_s^{V^*})(t_i e_i^{H^*} - r_i^* e_i^{V^*})P_2$$
$$+ (t_s e_s^{H^*} - r_s^* e_s^{V^*})(r_i e_i^{H^*} + t_i^* e_i^{V^*})P_3 \qquad (12)$$
$$+ (r_s e_s^{H^*} + t_s^* e_s^{V^*})(t_i e_i^{H^*} - r_i^* e_i^{V^*})P_3]e^{i\tilde{k}_p L}e^{i[\tilde{k}_p - (\tilde{k}_s + \tilde{k}_i)]z}$$

,

where further notation shortening was introduced via

$$P_1 = d_{22}(t_p^* e_p^V - r_p^* e_p^H) + d_{32}(t_p e_p^H + r_p e_p^V),$$
$$P_2 = d_{23}(t_p^* e_p^V - r_p^* e_p^H) + d_{33}(t_p e_p^H + r_p e_p^V), \qquad (13)$$
$$P_3 = d_{24}(t_p^* e_p^V - r_p^* e_p^H) + d_{34}(t_p e_p^H + r_p e_p^V).$$

To find the state, we have to substitute the components of the Jones vectors $e_{s,i}^{H,V}$ with the corresponding photon creation operators. In this case, transformations (7) and (10) are equivalent to the unitary transformations of a beamsplitter with two input and two output polarization modes. Substituting (12) into (2) and grouping the components with the same pair of the creation operators, we can finally find the two-photon polarization state in the qutrit form (1) with the complex amplitudes

$$C_1 = \sqrt{2} \int_{-L}^{0} dz [r_s r_i P_1 + t_s t_i P_2 + (t_s r_i + r_s t_i)P_3]e^{i[\tilde{k}_p - (\tilde{k}_s + \tilde{k}_i)]z},$$

$$C_2 = \int_{-L}^{0} dz [(r_s t_i^* + t_s^* r_i)P_1 - (t_s r_i^* + r_s^* t_i)P_2$$
$$+ (t_s t_i^* - r_s r_i^* + t_s^* t_i - r_s^* r_i)P_3]e^{i[\tilde{k}_p - (\tilde{k}_s + \tilde{k}_i)]z}, \qquad (14)$$

$$C_3 = \sqrt{2} \int_{-L}^{0} dz [t_s^* t_i^* P_1 + r_s^* r_i^* P_2 - (t_s^* r_i^* + r_s^* t_i^*)P_3]e^{i[\tilde{k}_p - (\tilde{k}_s + \tilde{k}_i)]z}.$$

The polarization state vector has to be further normalized with the norm $\sqrt{|C_1|^2 + |C_2|^2 + |C_3|^2}$. Although we use the normalized values of the complex amplitudes for the analysis of the two-photon polarization state (Extended Data Fig. 9), the norm itself shows the relative generation efficiency for different parameters of the LC, such as length and twist (Fig. 4c,d).

Further development of the model involves more strict quantum-optical calculations, with the real angular and spectral distributions of the generated photons, as well as the spatial properties of the pump beam, internal reflections of both the pump and the generated photons, and so on. Furthermore, the approximation of a non-depleted pump is valid only in the low-gain regime of SPDC, while such a source is incredibly promising for generating squeezed vacuum and twin beams. Finally, we assume a perfect uniform twist of the molecules, which is hard to achieve experimentally, especially for twists not multiple to π.

Although this model is significantly simplified, it proved to be reliable and provides a great insight into the physics of this type of material.

## Data availability

The data that support the findings of this study are available at https://doi.org/10.5281/zenodo.11072682 (ref. 52).

## Code availability

The code for analysis of the data is available at https://doi.org/10.5281/zenodo.11072682 (ref. 52).

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

**Acknowledgements** We thank Merck Electronics KGaA for providing the FNLC material, N. Osterman for suggesting the use of an FNLC as the nonlinear medium and J. Milivojević for assembling the LC cells. We acknowledge financial support from the European Research Council (ERC) under the European Union's Horizon 2020 research and innovation programme (grant agreement number 851143), from the Slovenian Research and Innovation Agency (ARIS) (P1-0099, P1-0192) and from Deutsche Forschungsgemeinschaft (429529648 - TRR 306 QuCoLiMa). The project/research is part of the Munich Quantum Valley, which is supported by the Bavarian state government with funds from the Hightech Agenda Bavaria. V.S. and M.V.C. are part of the Max Planck School of Photonics supported by BMBF, Max Planck Society and Fraunhofer Society.

**Author contributions** M.V.C. and M.H. conceived the idea and supervised the work. N.S. prepared the samples. V.S., A.K. and E.K. performed experiments and data analysis. V.S. and A.K. performed theoretical modelling and data representation. V.S., A.K., N.S., M.V.C. and M.H. worked on the text of the paper and the Supplementary Information.

**Funding** Open access funding provided by Max Planck Society.

**Competing interests** The authors declare no competing interests.

**Additional information**
**Correspondence and requests for materials** should be addressed to Maria V. Chekhova.

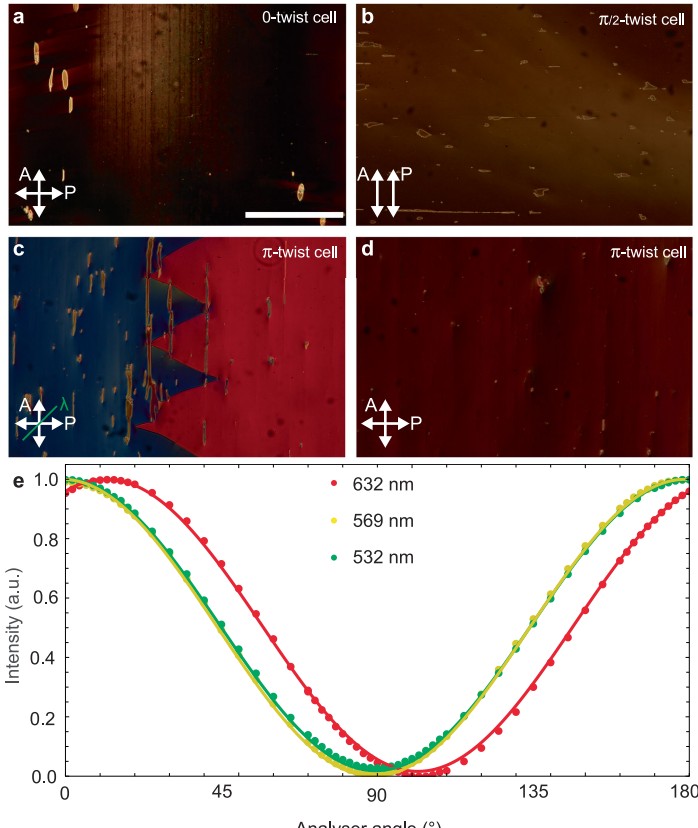

**Extended Data Fig. 1 | Polarization microscopy (POM) images of different 0-, π/2- and π-twisted cells. a)** A 0-twist cell as observed between crossed polarizers shows almost perfect extinction indicating uniform molecular orientation across the cell thickness. Scale bar, 500 μm. **b)** π/2-twist cells as observed between parallel polarizers. The imperfect extinction indicates a slight deviation of the molecular orientation direction across the cell from the ideal π/2 linear twist. Molecular orientation direction across the cell is analysed in **d. c)** A π-twisted cell as observed between crossed polarizers and a full-wave plate λ shows the division into two domains of opposite handedness (as evidenced by the sierra-walls), typical for antiparallel rubbed cells in which molecular orientation follows a π-twist structure across the sample thickness[5,53]. Elongated domains indicate surface defects, which are absent in the area of the electrode gap (**d**), where a single domain was imaged. **e)** Normalized transmitted intensity for a π/2-twist cell at different wavelengths (532 nm, 569 nm and 632 nm) as a function of the analyzer rotation where zero angle corresponds to the analyzer perpendicular to the polarizer. Circles denote experimental data acquired with the incoming polarization parallel to the bottom glass surface rubbing. Full lines correspond to transmission spectra simulations performed considering a linear twist structure for the molecular orientation. The best fits were obtained using 84° twist for 532 nm and 632 nm curves (with rotation from 5° to 89° from bottom to top surface, with respect to incoming polarisation) and 85° for 569 nm curve (rotation from 0° to 85°). The fact that no simple linear twist profile was found that simultaneously fits the three sets of data, indicates that the twist profile could deviate slightly from linear. This variation, together with the twist not being exactly 90°, could explain the differences in the calculated and measured polarization state in Fig. 4a at π/2 twist.

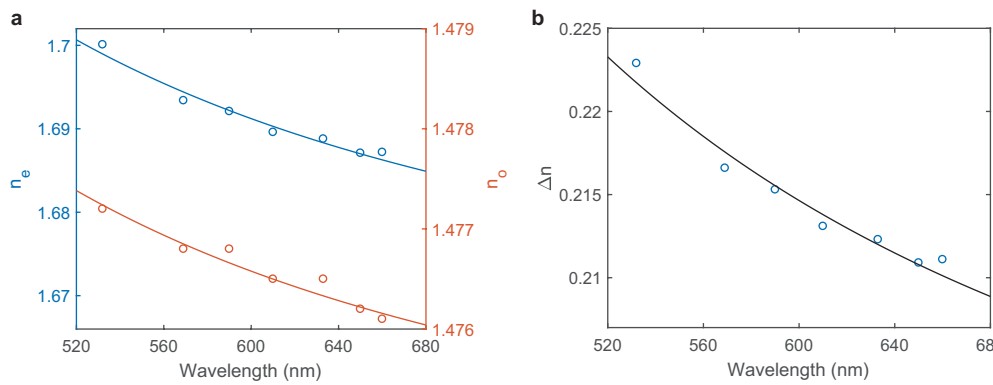

**Extended Data Fig. 2 | Refractive indices and birefringence of FNLC-1751 measured versus the wavelength. a)** Measured values and fitting functions for the extraordinary (blue) and ordinary (orange) refractive indices at different wavelengths. **b)** Values of birefringence extracted from data in panel **a**. In all cases, the fitting functions are two-term Cauchy models ($A + B/\lambda^2$). The resulting parameters are $A = 1.663$, $B = 10256$ nm$^2$ for the extraordinary refractive index and $A = 1.474$, $B = 874$ nm$^2$ for the ordinary refractive index.

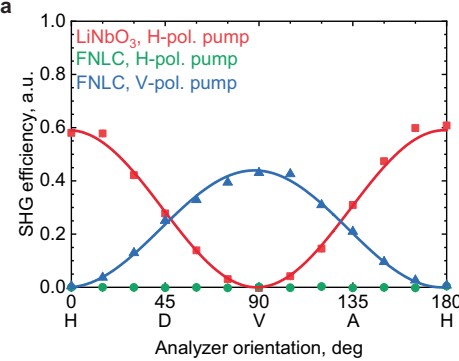
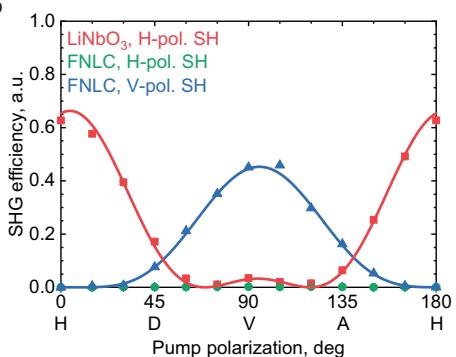

**Extended Data Fig. 3 | Measurement of the SHG efficiency in LC to determine the nonlinear tensor *d*. a)** We measured the SHG efficiency with a fixed pump polarization (horizontal or vertical) in LC (green and blue points, respectively) and compared it with the SHG efficiency in LN (red points). **b)** Similar measurements were done with the fixed orientation of the analyzer and a variable pump polarization. After fitting the results with the corresponding theoretical curves and comparing the SHG efficiencies in LN and LC, we concluded that out of six components of the *d* nonlinear tensor of LC that we were able to retrieve, only one is significant, $d_{33} \approx 20$ pm/V.

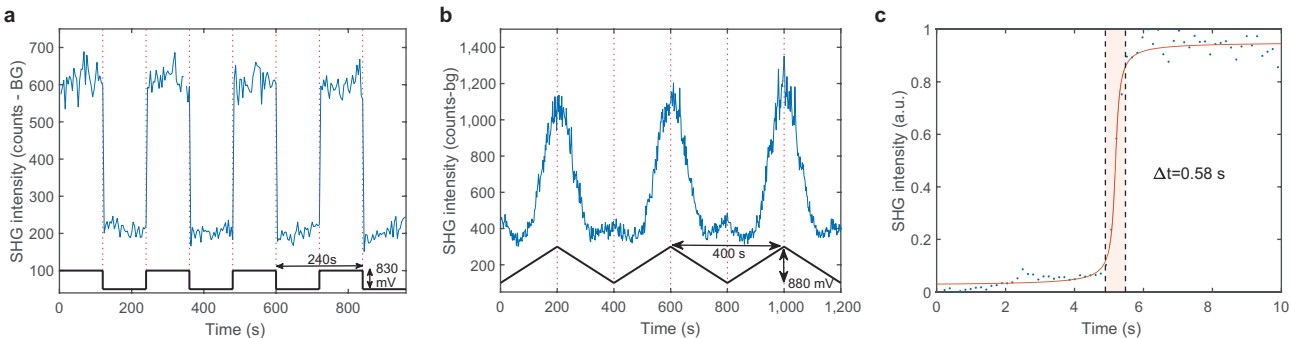

**Extended Data Fig. 4 | SHG switching under the applied electric field in the π-twisted LC. a)** Background-subtracted SHG intensity over time while applying square 830 mV voltage with 240 s period. We can see discrete transitions when the field is turned on or off and a stable behavior during the time the field remains fixed. The contrast between the on and off states of the field corresponds to a threefold increase in intensity. **b)** Background-subtracted SHG intensity over time while applying linearly increasing/decreasing 880 mV voltage pulses. The SHG intensity follows the voltage profile, the only exception being the low-voltage regions where the intensity is not really changing within the measurement error. We attribute this to the fact that electric field is insufficient to reorient the molecules and change the sample structure. The contrast between the on and off state is again threefold. **c)** Characteristic profile of the LC SHG intensity when the voltage is quickly switched on. The transition time of the molecular reorientation and relaxation into the new configuration driven by the applied field is around half a second (denoted by the shaded area). It is defined as the difference between the points where the amplitude is at 10% and 90%, respectively. The function fitted to the data is $a \cdot \mathrm{atan}((x-c)/b) + d$. In the presented case, 1 V was applied to the sample without the twist. No significant difference was observed for different field strengths or sample configurations (twist). The same behavior is expected also for SPDC, but it is harder to reliably measure with such temporal resolution due to much lower intensity.

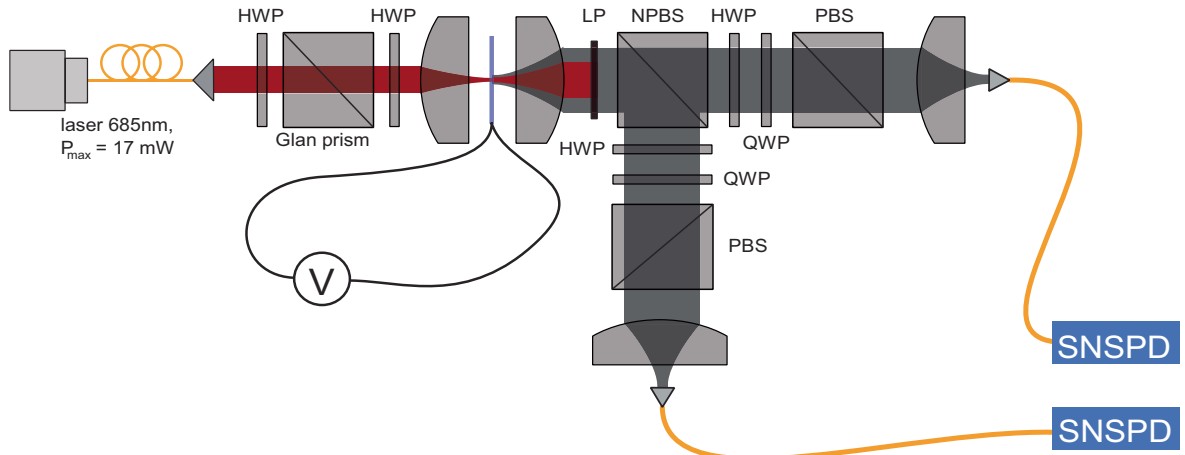

**Extended Data Fig. 5 | Experimental setup for generating and detecting entangled photons from LC.** After the power and polarization control, the pump (CW laser light @ 685 nm) was focused in the LC cell. Generated photon pairs were sent to the Hanbury Brown - Twiss setup and detected via coincidence measurements. HWP, half-wave plate; QWP, quarter-wave plate; LP, long-pass filter; NPBS, non-polarizing beam-splitter; PBS, polarizing beam-splitter; SNSPD, superconducting nanowire single-photon detector.

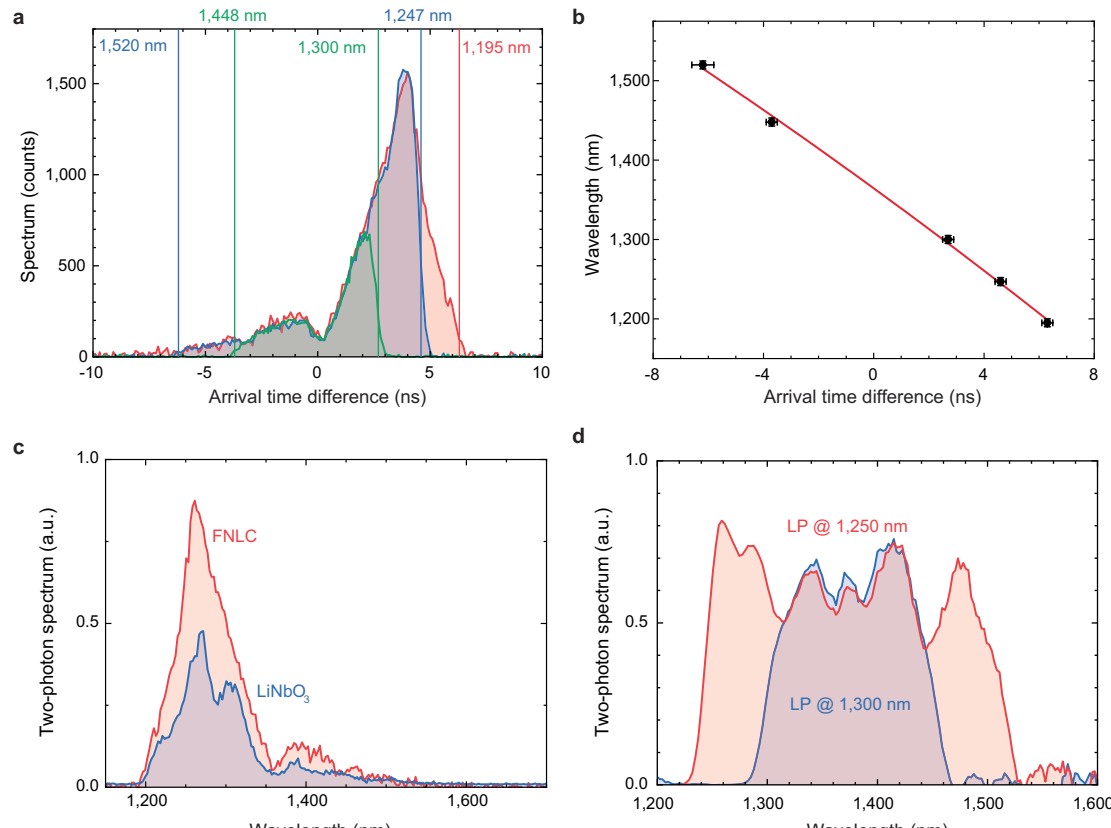

**Extended Data Fig. 6 | Two-photon spectrum measurement. a)** We measured the histogram of arrival time differences for several sets of frequency filters and matched their edges with the transmission spectra of the filters. **b)** The obtained points were fitted to map the detection time delay to the wavelength of the dispersed photon. **c)** We additionally measured the spectrum of 7 µm thick LN wafer with a known two-photon spectrum as a reference to account for the spectral losses in the setup. **d)** The final spectrum of photon pairs generated in LC was normalized to the reference spectrum. The measured spectrum is solely limited by the detection efficiency and the spectral filters used in the experiment. Due to the relaxed phase-matching, the generated two-photon spectrum should be much broader, occupying several octaves.

| Polarization state | HWP$_1$, deg | QWP$_1$, deg | HWP$_2$, deg | QWP$_2$, deg |
|:---:|:---:|:---:|:---:|:---:|
| H-H | 0 | 0 | 0 | 0 |
| H-V | 0 | 0 | 45 | 0 |
| V-V | 45 | 0 | 45 | 0 |
| H-D | 0 | 0 | 22.5 | 45 |
| H-R | 0 | 0 | 0 | −45 |
| V-A | 45 | 0 | −22.5 | 45 |
| V-L | 45 | 0 | 0 | 45 |
| D-D | 22.5 | 45 | 22.5 | 45 |
| D-R | 22.5 | 45 | 0 | −45 |

**Extended Data Fig. 7 | The protocol for two-photon polarization tomography of a qutrit state.** Indices 1 and 2 correspond to the different outputs of the NPBS.

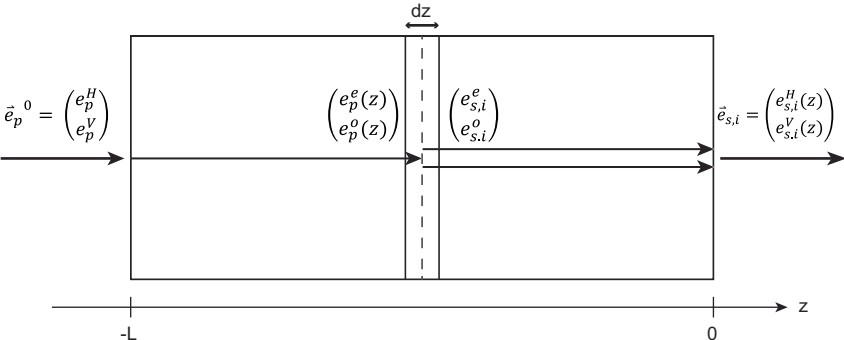

**Extended Data Fig. 8 | Notation used to describe the propagation of the pump and the generated photon pairs within the sample in the theoretical model.** The input Jones vector is converted into the local frame of reference (o,e); then its evolution along the sample is described, and finally it is converted back to the laboratory frame of reference H,V.

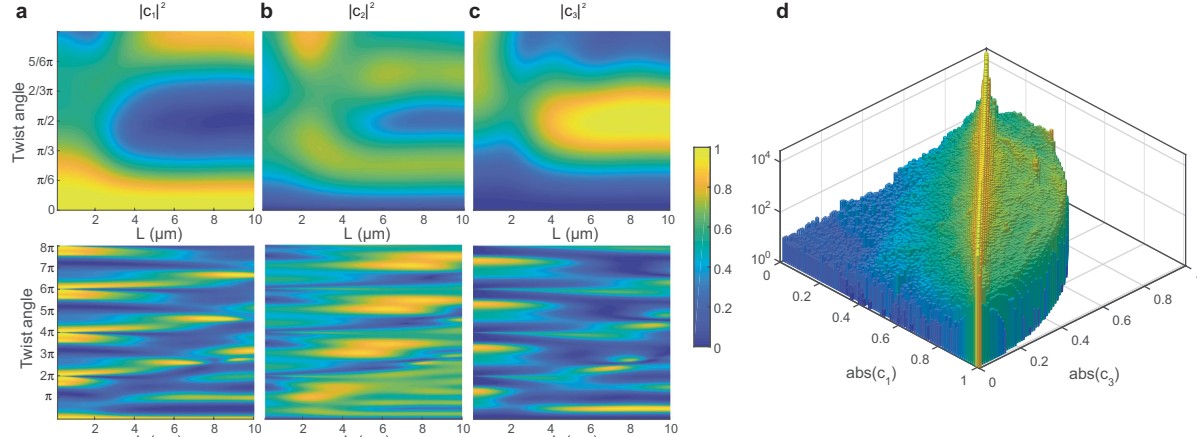

**Extended Data Fig. 9 | Calculated polarization states of photon pairs.** The squared amplitudes represent the probabilities for each of the basis states. **(a-c)** Values of $|c_1|^2$, $|c_2|^2$, and $|c_3|^2$ versus the sample thickness and the twist angle. Top panels present the region related to the current experiments, while the bottom panels show the effect of larger twist angles, up to 4 full twists, demonstrating the possibility to generate broader sets of polarization states. In all calculations, the pump polarization was diagonal (*D*). **d)** Theoretical prediction on the possible states that can be produced. All possible combinations of $|c_1|$ and $|c_3|$ that can be achieved through varying the physical parameters of the sample (namely, sample length and twist angle) and pump polarization are presented. Different combinations are obtained with different occurrence rate, as certain combinations will only be satisfied by very specific set of parameters. However, this result shows that a set of parameters that can generate any desired combination always exists. Sample length up to 40 μm and twist angle up to $8\pi$ is used in the simulation as well as 6 different pump polarizations, namely *H*, *V*, *A*, *D*, *R* and *L*.