## [Peer Review File · Nature]

Manuscript Title: Tuneable entangled photon pair generation in a liquid crystal

Reviewer Comments & Author Rebuttals

Reviewer Reports on the Initial Version:

Referees' comments:

Referee #1 (Remarks to the Author):

The article is very well written, with comprehensive review of current state-of-the-art practice in entangled photon pair generation, detailed description of methodology and analyses of observed effects/data. The results in FNLC [in conjunction with a Lithium Niobate crystalline thin film] are in reasonable agreement with theoretical expectation, and supporting the authors' claim that "this is the first time reconfigurable/tunable entangled photons production in a liquid crystals or organic materials" is achieved. Experimental results and discussions are supported by detailed theories on various aspects of FNLC optical characteristics [polarizations, crystalline alignment, and reorientation, ...etc.] in the Supplement.

Comments

- Materials that possess high 2nd order nonlinearity such as 2HG crystals used in almost all laser systems and nonlinear frequency mixing operations are highly efficient because the cross-section of $w_1+w_2 \rightarrow w_3$ is high [due to sizeable nonlinear coefficient d_{ij}] simply because the frequencies w_1 and w_2 are fixed [by the incident light]. On the other hand, the reverse process of a particular spontaneous down conversion process [$w_3 \rightarrow w_1+w_2$] is but one of the almost infinite number of possible combination for possibilities for w_1+w_2]; the cross-section for generating a particular entangled paired photons is therefore very small, though detectable by very sensitive experimental set up.

BUT will such process eventually rises up to being a practical and efficient means of generating entangled photon pairs? If so, how and what are the steps/advances needed to make the process more efficient [with phase-matching and coherent beam coupling....]?

- As clearly illustrated previously by many others and the authors, materials with high second order susceptibilities or nonlinear coefficients d_{ij} 's allow observation of entangled paired photons generation. In this case, the FNLC sample thickness is smaller than the coherent length [to avoid phase mismatch that will lead to lower conversion efficiency]

While it opens new future research possibilities, it remains to be proven that FNLC is an efficient and robust platform that will "outperform standard nonlinear optical materials in terms of functionality, brightness and the tunability of the generated quantum state. The concepts developed here can be extended to complex topological structures, macroscopic devices, and multipixel tunable quantum light sources" [quoting from the abstract].

To support such claim, the authors should provide an [even if optimistic] estimate of the actual efficiency for a particular paired entangled photon generation, or quote the theoretical estimate if known or reported previously or in this article, as a function of the nonlinear coefficient and process

geometry involving FNLC [e.g. the geometry depicted in Figure 1 or thin film (in-plane or normal to pane) in free space or waveguide (fiber or plane)].

- In general, a polarized laser of 6 mW or so focused to a spot size of ~ 5 microns puts out an intensity $I \sim 24000$ Watts/cm² would and will inevitably create several complications and may not be conducive to entangled photon pair generation, including (i) Molecular reorientations, (ii) heating effects on ITO electrode, (iii) other nonlinear optical effects such as stimulated orientational scattering leading to cross-polarization beam coupling, and (iv) photorefractivity [beam coupling], ...in liquid crystals.

Did the authors observe any of the effects? If not, please explain why in their FNLC samples, these effects are not present or avoided.

If the authors observed any of the nonlinear optical effects, explain how they affect or do not affect photon pair production, e.g. in stimulated polarization scattering as an example.

Referee #2 (Remarks to the Author):

This manuscript reports the implementation of the spontaneous parametric downconversion (SPDC) process in ferroelectric nematic liquid crystal and the subsequent demonstration of a source of entangled photon pairs in the polarization degree of freedom. One of the most important results is the high degree of tuning of the generated state regarding the polarization between the photons comprising a pair. State tuning is achieved by applying an electric field (a few volts) or twisting the molecular orientation along the sample. Beyond the polarization state, the photons in a pair are indistinguishable in spatial mode and spectrum.

The reported work is undoubtedly original; there are no reports in the literature of SPDC sources based on liquid crystals or sources with such broad flexibility for controlling the quantum entanglement properties of the two-photon state generated by SPDC in the same medium with second-order nonlinearity.

The description of the research methodology and the obtained results is clear, concise, and well-justified. Every aspect related to the author's novel SPDC source proposal is considered, justified, and contrasted with already reported alternative sources: sample preparation, measurement of linear and non-linear optical properties, characterization of SPDC source properties, such as the correlation between the detection times of the two photons (rate of real and accidental coincidences), emission spectrum, reconstruction of the two-photon polarization density matrix. Authors validate their experimental results with a theoretical model, which, although under certain approximations, provides results that agree well with the experimental data. It is worth highlighting the completeness of the work developed and reported, which helps in the assimilation of results and the proposal, leaving no room for doubt regarding the quality and impact of the work.

Beyond the scope of the reported work, the authors describe some future perspectives for their proposal, which are well supported in the manuscript or the supplementary material.

The presentation of the experimental data, such as those derived from the theoretical model, is

appropriate. Slight discrepancies between the experimental and theoretical data were obtained. Still, the reasons were clarified in detail, generally attributed to the restrictions imposed on the theoretical model or possible defects in manufacturing the samples.

Throughout the manuscript, credits are given for previous works.

I have no doubts about the novelty and impact of the work. The manuscript is clear and coherent, and I did not detect any inconsistency among the different sections and the extended and supplementary material.

The work presented can be of great interest to the community dedicated to quantum optics and quantum information; in fact, the work can become a watershed for a new generation of sources of entangled two-photon states based on SPDC. Therefore, the manuscript is appropriate for publication in Nature. However, I suggest that the following points be taken into account:

The authors presented their proposal as a new paradigm for sources of quantum states with tunable entanglement properties. In such a case, it is desirable to discuss the compatibility of this kind of source with other devices needed to implement any quantum technology based on entangled two-photon states. It is well known that in the process of developing some quantum technology, the aspects of scalability and integrability must be considered.

Although the manuscript mentions that the emission rates of the proposed source are comparable to those of other SPDC-based schemes, there is always the issue of how to incorporate the source into a circuit, integrated or not.

And in the same direction, is the cost/benefit ratio of the proposal significant?

On the other hand, the unique feature of the proposal presented by the authors is the degree of tuning of the quantum state, which is well contrasted with SPDC sources implemented in media, such as bulk crystals, integrated waveguides, and QPM materials. However, contrast is not made with media exhibiting third-order nonlinearity. Since spontaneous four-wave mixing (SFWM) is the process in competition with SPDC-based sources, such a comparison would be appropriate. There have been demonstrations of SFWM sources with enough flexibility in engineering the properties of the resulting two-photon states in a single medium.

Some minor details, such as the mention of sections 6 and 7 of the supplementary material, which do not exist, and a no defined λ letter in Extended data Fig. 1, panel c. Still, I assume all these will be considered at the editing stage if the manuscript is accepted for publication.

Author Rebuttals to Initial Comments:

Response to reviewers

We would like to thank both referees for carefully reading our manuscript. Below, we address the questions and comments from both of them. Our replies are in blue font, and the changes made in the manuscript are in red font.

Referee #1 (Remarks to the Author):

The article is very well written, with comprehensive review of current state-of-the-art practice in entangled photon pair generation, detailed description of methodology and analyses of observed effects/data. The results in FNLC [in conjunction with a Lithium Niobate crystalline thin film] are in reasonable agreement with theoretical expectation, and supporting the authors' claim that "this is the first time reconfigurable/tunable entangled photons production in a liquid crystals or organic materials" is achieved. Experimental results and discussions are supported by detailed theories on various aspects of FNLC optical characteristics [polarizations, crystalline alignment, and reorientation, ...etc.] in the Supplement.

We thank the reviewer for the positive opinion about our article and the detailed comments.

Comments

- Materials that possess high 2nd order nonlinearity such as 2HG crystals used in almost all laser systems and nonlinear frequency mixing operations are highly efficient because the cross-section of $w_1+w_2 \rightarrow w_3$ is high [due to sizeable nonlinear coefficient d_{ij}] simply because the frequencies w_1 and w_2 are fixed [by the incident light]. On the other hand, the reverse process of a particular spontaneous down conversion process [$w_3 \rightarrow w_1+w_2$] is but one of the almost infinite number of possible combination for possibilities for w_1+w_2 ; the cross-section for generating a particular entangled paired photons is therefore very small, though detectable by very sensitive experimental set up.

BUT will such process eventually rises up to being a practical and efficient means of generating entangled photon pairs? If so, how and what are the steps/advances needed to make the process more efficient [with phase-matching and coherent beam coupling....]?

We would like to stress that, generally, SPDC can be very efficient provided that the second-order susceptibility is high, the material is long and phase-matched, and the pump is strong. As an example, up to 30% of the pump was converted in a 5 mm-long lithium niobate [<https://opg.optica.org/ol/abstract.cfm?uri=ol-45-15-4264>]. In this work, we show that the susceptibility is comparable with that of lithium niobate, the length can be a few mm if the molecular twist is used for quasi-phasematching, and the pump can be strong: we did not observe any damage or other nonlinear effects by focused pump radiation (see our comments below).

Therefore, nothing makes ferroelectric nematic liquid crystal (FNLC) less suitable for SPDC than other nonlinear crystals. But here, we additionally show the advantages it offers: tunability of the polarization state and efficiency under electric field and molecular twist. We have added the following text:

Due to their nonlinear coefficient comparable to the best nonlinear crystals, such as lithium niobate, and high damage threshold, FNLCs are perfectly suitable for practical applications. Furthermore, high-quality liquid crystal devices such as liquid crystal displays (LCDs) are made on an industrial scale, which, combined with our work, opens a path to scalable and cheap production of quantum light sources while at the same time exceeding the existing ones in efficiency and functionality.

- As clearly illustrated previously by many others and the authors, materials with high second order susceptibilities or nonlinear coefficients d_{ij} 's allow observation of entangled paired photons generation. In this case, the FNLC sample thickness is smaller than the coherent length [to avoid phase mismatch that will lead to lower conversion efficiency]

While it opens new future research possibilities, it remains to be proven that FNLC is an efficient and robust platform that will “outperform standard nonlinear optical materials in terms of functionality, brightness and the tunability of the generated quantum state. The concepts developed here can be extended to complex topological structures, macroscopic devices, and multipixel tunable quantum light sources” [quoting from the abstract].

To support such claim, the authors should provide an [even if optimistic] estimate of the actual efficiency for a particular paired entangled photon generation, or quote the theoretical estimate if known or reported previously or in this article, as a function of the nonlinear coefficient and process geometry involving FNLC [e.g. the geometry depicted in Figure 1 or thin film (in-plane or normal to pane) in free space or waveguide (fiber or plane)].

The experimentally detected photon pair rate is 1.4 kHz with 8 mW pump in a 8 μm layer (Fig. 1c), which, to the best of our knowledge, already exceeds SPDC rates reported for most micrometer-thickness materials. As we show in the last part of the paper, the length of the FNLC can be increased by using the molecular twist with the right pitch. This way, SPDC is quasi-phasematched, and the coincidence rate scales quadratically with the length. Then, for a 200 μm sample, according to Fig. 4d, we expect a rate of 1 MHz, which is not only enough for quantum technological applications but too high for most single-photon detectors.

We modified the sentence about quasi-phase matching to make the estimate more convincing:

In this case, a 200 μm sample will generate 625 times more pairs than the current sample (Fig. 4d), reaching count rates of almost 1 MHz, more than enough for most quantum technological applications.

- In general, a polarized laser of 6 mW or so focused to a spot size of ~ 5 microns puts out an intensity $I \sim 24000$ Watts/cm² would and will inevitably create several complications and may not be conducive to entangled photon pair generation, including (i) Molecular reorientations, (ii) heating effects on ITO electrode, (iii) other nonlinear optical effects such as stimulated orientational scattering leading to cross-polarization beam coupling, and (iv) photorefractivity [beam coupling],in liquid crystals.

Did the authors observe any of the effects? If not, please explain why in their FNLC samples, these effects are not present or avoided.

If the authors observed any of the nonlinear optical effects, explain how they affect or do not affect photon pair production, e.g. in stimulated polarization scattering as an example.

We agree with the reviewer that the effects of the pump beam on the sample are very important and need to be considered carefully. As discussed below, there are several indications that there are no other significant nonlinear effects present in our samples.

1. The most substantial indication is the linear increase of the coincidence rate with the pump power (Fig. 1c). If there had been significant heating, for example, transition into some other LC phase, there would be a steep decrease in the coincidence rate at a certain pump power. Similarly, the reorientation of molecules would lead to a nonlinear response in the coincidence rate versus the pump power. We do not observe any such effect.
2. Further, there is no significant molecular reorientation since the experimentally measured polarization state matches well with the calculated one.
3. In the brightfield and crossed polarizer microscopy images we do not observe any changes on the sample (both momentary or permanent) caused by the pump beam. Even several

hours of illumination with a few mW did not cause any visible permanent changes. Only above 60 mW and illumination times of more than 30 min caused a small permanent spot.

4. According to the literature (<https://journals.aps.org/pre/abstract/10.1103/PhysRevE.73.021705>), the reorientation of LC molecules (optical Fréedericksz transition) in nematic LC samples happens at 30 mW of laser power for a diffraction-limited spot through a high NA objective, which is well above our pump intensity. In FNLC, the threshold for optical Fréedericksz transition could be different than in nematic, but as described in the previous points, this would be visible in the images or in the change of the photon pair rate or state.
5. In our case, we illuminate the space between the electrodes, where there is no ITO. Even if illuminating the ITO, it takes several tens of mW of laser power to bring the temperature of 5CB above its melting point at ~ 35 °C (<https://journals.aps.org/pre/abstract/10.1103/PhysRevE.76.051406>). In comparison, the FNLC used here has a transition temperature to the M2 phase at 45 °C. Also, the refractive index (and birefringence) of FNLCs change relatively little with temperature (<https://arxiv.org/abs/2401.09675>).
6. Ferroelectric LCs are indeed known to have significant photorefractivity, but these are mainly smectic chiral LCs (SmC*) (<https://pubs.acs.org/doi/full/10.1021/jp030456d>). However, for the photorefractive effect to occur, two strong beams are necessary. While there indeed are separated signal and idler beams, they are certainly not strong enough to produce any relevant beam coupling through material nonlinearities.
7. Other nonlinear effects that could, in principle, occur in this sort of materials, such as four-wave mixing processes, optical phase conjugation, as well as stimulated orientational scattering, are, however, much weaker than SPDC since they are governed by third-order nonlinearity (χ^3), while SPDC depends on second-order nonlinearity (χ^2). These effects become relevant at much higher powers achieved only under pulsed radiation, but this was not the case in our experiments. If, however, higher-order nonlinear effects appear under pulsed pumping, the radiation resulting from them is usually in a different part of the spectrum and would be filtered out by the filters used in our setup in the same way as we filtered the pump from the SPDC signal.
8. Studies confirm that stimulated orientational scattering occurs in certain nematic liquid crystals, such as E-7, under illumination by a CW pump of relatively low power (<https://journals.aps.org/pre/abstract/10.1103/PhysRevE.62.6722>, <https://journals.aps.org/pre/abstract/10.1103/PhysRevE.55.5603>). The effect scales exponentially with the thickness of a liquid crystal and was observed from the 100- μm thick samples (and thicker). On the other hand, we have sub-10 μm samples and a 10 mW pump.

We have added a new section to Supplementary Information (Section 3) describing that we do not observe any other nonlinear effects. We have also added a sentence to the main text:

At this pump power, we did not observe any other nonlinear effects or permanent damage to the sample (Section 3 of Supplementary Information).

Referee #2 (Remarks to the Author):

This manuscript reports the implementation of the spontaneous parametric downconversion (SPDC) process in ferroelectric nematic liquid crystal and the subsequent demonstration of a source of entangled photon pairs in the polarization degree of freedom. One of the most important results is the high degree of tuning of the generated state regarding the polarization between the photons comprising a pair. State tuning is achieved by applying an electric field (a few volts) or twisting the molecular orientation along the sample. Beyond the polarization state, the photons in a pair are indistinguishable in spatial mode and spectrum.

The reported work is undoubtedly original; there are no reports in the literature of SPDC sources based on liquid crystals or sources with such broad flexibility for controlling the quantum entanglement properties of the two-photon state generated by SPDC in the same medium with second-order nonlinearity.

The description of the research methodology and the obtained results is clear, concise, and well-justified. Every aspect related to the author's novel SPDC source proposal is considered, justified, and contrasted with already reported alternative sources: sample preparation, measurement of linear and non-linear optical properties, characterization of SPDC source properties, such as the correlation between the detection times of the two photons (rate of real and accidental coincidences), emission spectrum, reconstruction of the two-photon polarization density matrix. Authors validate their experimental results with a theoretical model, which, although under certain approximations, provides results that agree well with the experimental data. It is worth highlighting the completeness of the work developed and reported, which helps in the assimilation of results and the proposal, leaving no room for doubt regarding the quality and impact of the work.

Beyond the scope of the reported work, the authors describe some future perspectives for their proposal, which are well supported in the manuscript or the supplementary material.

The presentation of the experimental data, such as those derived from the theoretical model, is appropriate. Slight discrepancies between the experimental and theoretical data were obtained. Still, the reasons were clarified in detail, generally attributed to the restrictions imposed on the theoretical model or possible defects in manufacturing the samples.

Throughout the manuscript, credits are given for previous works.

I have no doubts about the novelty and impact of the work. The manuscript is clear and coherent, and I did not detect any inconsistency among the different sections and the extended and supplementary material.

The work presented can be of great interest to the community dedicated to quantum optics and quantum information; in fact, the work can become a watershed for a new generation of sources of entangled two-photon states based on SPDC. Therefore, the manuscript is appropriate for publication in Nature. However, I suggest that the following points be taken into account:

We thank the reviewer for emphasizing the novelty and impact of our work and for the comments.

The authors presented their proposal as a new paradigm for sources of quantum states with tunable entanglement properties. In such a case, it is desirable to discuss the compatibility of this kind of source with other devices needed to implement any quantum technology based on entangled two-photon states. It is well known that in the process of developing some quantum technology, the aspects of scalability and integrability must be considered.

Although the manuscript mentions that the emission rates of the proposed source are comparable to those of other SPDC-based schemes, there is always the issue of how to incorporate the source into a circuit, integrated or not.

We agree with the reviewer that the integration of these sources is an important consideration. This work is a proof of principle for photon pair generation in a liquid crystal. Further development might imply the integrability of this platform with existing optical platforms, for instance, by filling a hollow-core fiber with a liquid crystal (<https://pubs.aip.org/aip/apl/article/85/12/2181/329555>) or by combining a liquid crystal with a waveguide (<https://opg.optica.org/jlt/abstract.cfm?uri=jlt-38-15-4045>), or a metasurface (<https://opg.optica.org/jlt/abstract.cfm?uri=jlt-38-15-4045>). We emphasize the integrability by adding these references to the manuscript:

The liquid nature of FNLCs enables their integration with existing optical platforms such as fibers [41], waveguides [42], and metasurfaces [43].

And in the same direction, is the cost/benefit ratio of the proposal significant?

We thank the reviewer for this question. Indeed, these materials are low-cost, low-maintenance, really robust and stable. Liquid crystals can be produced in large quantities cheaply and are employed on an industrial scale in devices such as spatial light modulators (SLMs) and liquid crystal displays (LCDs). These established methods can be transferred to the manufacturing of nonlinear optical devices. We modified the following sentence in the manuscript:

Furthermore, high-quality liquid crystal devices such as liquid crystal displays (LCDs) are made on an industrial scale, which, combined with our work, opens a path to scalable and cheap production of quantum light sources while exceeding the existing ones in efficiency and functionality.

On the other hand, the unique feature of the proposal presented by the authors is the degree of tuning of the quantum state, which is well contrasted with SPDC sources implemented in media, such as bulk crystals, integrated waveguides, and QPM materials. However, contrast is not made with media exhibiting third-order nonlinearity. Since spontaneous four-wave mixing (SFWM) is the process in competition with SPDC-based sources, such a comparison would be appropriate. There have been demonstrations of SFWM sources with enough flexibility in engineering the properties of the resulting two-photon states in a single medium.

In this work, we focus on the tunability of the polarization two-photon state. In both conventional SPDC- and SFWM-based sources, the polarization state is strictly defined by the given nonlinear tensor (second or third-order, respectively) of the source. To the knowledge of the authors, there is no possibility of changing the two-photon polarization state in any SFWM source without re-designing the source [<https://www.nature.com/articles/s41598-017-06010-8>, <https://opg.optica.org/oe/fulltext.cfm?uri=oe-16-8-5721&id=157129>]. We emphasize this point in the manuscript:

Such performance and the possibility to dynamically control the two-photon state are superior to existing crystal SPDC or fiber SFWM sources. The latter are less efficient, require a strong pulsed pump and a relatively long nonlinear medium, and have no polarization tunability without re-designing the source [37,38].

Therefore, a nonlinear liquid crystal outperforms both conventional platforms. Moreover, the frequency tunability of SFWM is even more limited [<https://iopscience.iop.org/article/10.1088/2058-9565/aa7a37>, <https://opg.optica.org/ol/abstract.cfm?uri=ol-46-16-4033>] than that of SPDC in bulk crystals.

Some minor details, such as the mention of sections 6 and 7 of the supplementary material, which do not exist, and a no defined λ letter in Extended data Fig. 1, panel c. Still, I assume all these will be considered at the editing stage if the manuscript is accepted for publication.

We thank the reviewer for pointing out the discrepancies in the manuscript. For the reference to the sections containing the details of the experiment, we refer generally to the Methods.

In the Extended data Fig. 1c, λ indicates the full waveplate, which is rotated by 45 degrees to the polarizer and analyzer. A description was added to the caption.

Reviewer Reports on the First Revision:

Referees' comments:

Referee #1 (Remarks to the Author):

I have read the author(s)'s point-by-point answers have adequately addressed my comments on the efficiency, possibility of scaling up in interaction length by phase-matching, and most importantly, possible complications arising from nonlinear processes associated with focused milliwatt laser on liquid crystals [especially ferroelectric nematic]. In the latter context, the authors mentioned that the laser power used is below the optical 'Freedericksz transition' field [quoting ref. Phys. Rev. E 73, 021705], so they did not observe nonlinear optical field induced reorientation and no complication in their polarization dependence studies. On this point, I would like to point out that in that reference, the sample under study is "small colloidal particles in a nematic liquid crystal", and it is very different from a pristine FNLC - so the comparison is not appropriate. Nevertheless, it seems the authors have not observed any nonlinear optical field induced changes in FNLC, and that added statement in the revised manuscript is acceptable to me.

IC Khoo

Referee #2 (Remarks to the Author):

The authors have addressed the questions and observations made by both referees. They give convincing explanations (supported by literature in most cases) for each point. The current version of the manuscript (in terms of scientific content) is ready for publication in Nature. The work constitutes a valuable contribution to the state of the art. I do not have any further comments.